# Shark movements between islands in the Revillagigedo Archipelago and connectivity to other islands in the Eastern Tropical Pacific

Frida Lara-Lizardi[1,2,3], James T. Ketchum[2,3,4]\*, Alex R. Hearn[3,5], A. Peter Klimley[3,6], Felipe Galván-Magaña[1], Alex Antoniou[7], Randall Arauz[3,8,9], Sandra Bessudo[3,10], Eleazar Castro[11], Elpis J. Chávez[3,9], Eric E.G. Clua[12,13], Eduardo Espinoza[3,14], Chris Fischer[15], César Peñaherrera-Palma[3], Todd Steiner[3,16], Mauricio Hoyos-Padilla[2,3,7]\*

1 Instituto Politécnico Nacional, Centro Interdisciplinario de Ciencias Marinas, La Paz, Baja California Sur, México, 2 Pelagios Kakunjá, La Paz, Baja California Sur, México, 3 MigraMar, Bodega Bay, California, United States of America, 4 Centro de Investigaciones Biológicas del Noroeste-CIBNOR, Km 1 Carretera a San Juan de La Costa, La Paz, Baja California Sur, México, 5 Galapagos Science Center, Universidad San Francisco de Quito USFQ, Quito, Ecuador, 6 Department of Wildlife, Fish, and Conservation Biology, University of California, Davis, California, United States of America, 7 Fins Attached Marine Research and Conservation, Colorado Springs, Colorado, United States of America, 8 Marine Watch International. San Francisco, California, United States of America, 9 Centro Rescate de Especies Marinas Amenazadas CREMA, San José, Costa Rica, 10 Fundación Malpelo, Bogotá, Colombia, 11 Centro Interdisciplinario en Ciencias Aplicadas de Baja California Sur, La Paz, Baja California Sur, México, 12 Paris Science et Lettre (PSL), Ecole Pratique des Hautes Etudes, Centre de Recherche Insulaire et Observatoire de l'Environnement, Perpignan, France, 13 Labex Corail, Centre de Recherche Insulaire et Observatoire de l'Environnement, Opunohu, Moorea, French Polynesia, 14 Parque Nacional Galápagos, Isla Santa Cruz, Ecuador, 15 Ocearch, Park City, Utah, United States of America, 16 Turtle Island Restoration Network, Olema, California, United States of America

\* james@pelagioskakunja.org (JTK), mauricio@pelagioskakunja.org (MH-P)

## Abstract

There is a need to understand the degree to which sharks move between islands in Marine Protected Areas (MPAs) of the Eastern Tropical Pacific (ETP). Exposure to fishing activities becomes significant when no-take zones do not cover the critical areas that sharks use. We analyzed an ultrasonic telemetry dataset to assess how Galapagos sharks (*Carcharhinus galapagensis*) and silky sharks (*Carcharhinus falciformis*) move between the islands that comprise the Revillagigedo Archipelago (RA) and how they migrate to other islands in the ETP. In total, 92 sharks of both species were tracked from January 2010 to December 2018 in the region. Particularly, 39 sharks were detected in the Revillagigedo Archipelago (RA). Of these, 27 were resident at one island (behavior type I), 10 moved between two or more islands within a MPA (type II), and 3 sharks moved between MPAs (behavior type III): a silky shark tagged at Roca Partida (RA) that moved to Clipperton Atoll (CA), another silky shark moved from Wolf, Galapagos Archipelago (GA) to CA and back again and a Galapagos shark tagged at Socorro Island (RA), detected at CA, and finally recorded in Darwin Island (GA). This excursion was one of the longest movements ever recorded for the species (3,160 km). The long-distance dispersal observed in these two

**Data availability statement:** All relevant data are within the paper and its Supporting Information files.

**Funding:** The authors (MH-P and JTK) would like to thank the International Community Foundation, Ocearch, Chris Fischer Productions and National Geographic for providing the initial funding to tag many sharks and set up the acoustic receiver array in the Revillagigedo Archipelago. We are gratiefiul to Club Cantamar, Fins Attached (Alex Antoniou), Sharks Mission France and Quino El Guardian for supporting our expeditions to the islands. We also thank the Alliances WWF-Telmex-Telcel and WWF-Fundación Carlos Slim and Ocean Blue Tree for additional support for this project.

**Competing interests:** NO authors have competing interests.

species underscores the necessity for international collaboration. Such cooperation is essential to implement effective shark protection measures, including swimways or *MigraVías*, and other conservation tools in the ETP region.

## Introduction

Apex marine predators often congregate at specific sites at islands and seamounts, referred to as biological hotspots [1], where they either rest [2] or engage in social activities between foraging sessions—a social system often termed central-place foraging [3]. Understanding shark and marine predator movements is essential to: (a) evaluate hotspot interconnectedness in archipelagos [4]; (b) delineate movement corridors between hotspots [5]; and (c) establish or expand MPAs to improve migratory species' conservation and management [6].

While MPAs are crucial for safeguarding critical habitats and maintaining ecosystem connectivity by restricting human activities [6,7], migratory species frequently move beyond these protected zones, exposing them to fishing pressures in unprotected waters. A combined strategy, with fisheries managing measures such as, Temporal fishing bans, Total Allowable Catch and Total Allowable Effort, ensures population sustainability and allows for flexibility to adjust to shifting species movements driven by environmental changes, enhancing ecosystem resilience and better managing the ecological roles of migratory predators [7–10].

MPAs are areas where human activity is restricted for conservation purposes, typically aimed at safeguarding natural or cultural resources [6]. The functional and physical connections between various habitats, termed connectivity, are vital for preserving the biodiversity and resilience of an ecosystem [7]. Understanding movement pathways in an area can assist in: (a) informing management plans aimed at maintaining or restoring connectivity [8]; (b) enhancing the design and efficacy of MPAs [4]; and (c) defining the functional role of a diverse array of predators in marine ecosystems [7,10].

Many MPAs have been designated around oceanic islands designed to protect pelagic and highly migratory species, such as sharks. Among notable MPAs in the ETP (the region west of Mexico, Central and South America, between the tip of the Baja California Peninsula on the north and Peru on the south, and as far west as 150° longitude; [9–10]) are the following (in decreasing order in size): Revillagigedo National Park (RA; 148,000 km$^2$; [11]), Galapagos Marine Reserve (GA; 133,000 km$^2$; [12]), Cocos Island National Park (CO; 54,844 km$^2$; [13]), and Malpelo Island Flora and Fauna Sanctuary (MA; 20,959 km$^2$; [14]). More recently, the territorial waters of Clipperton Atoll (CA; 1,807 km$^2$ including the lagoon) was proposed as an MPA in 2016 [15].

These areas not only contribute to the protection of species with high ecological value, they provide habitat for endangered species and are of paramount cultural value [9,16]. Therefore, all the MPAs mentioned above have been designated as United Nations Educational, Scientific and Cultural Organization (UNESCO) World

Natural Heritage Sites [16]. UNESCO first recognized CO in 1997, then the GA in 2001, MA in 2006, and the RA in 2017 [11]. The latter is a large-scale MPA and was expanded from 6,366 km$^2$ to 148,000 km$^2$ in 2017, based on the knowledge of inter-insular connectivity and the large-scale movements of different shark species [11], creating the largest no-take zone in North America [17,18].

There is currently some evidence of inter-island movements of sharks in the ETP. Sharks may use islands as "stepping-stones" for long-distance oceanic dispersal [4,17–22]. For example, scalloped hammerhead sharks (*Sphyrna lewini)* at Wolf and Darwin islands in the GA moved over 100 km to Roca Redonda and Seymour Norte within the MPA, and others made longer-distance movements across the ETP to other isolated islands, such as CO (690 km) and MA (1,100 km) [4–5]. These inter-island movements may be driven by the need to exploit resources and suitable environmental conditions at different life stages, ensuring their survival and reproductive success. However, these movements in and out of MPAs imply that scalloped hammerheads are vulnerable to both domestic and multinational fisheries in the high seas [4,18,20]. Regular movements across MPA boundaries highlight the need for cooperation between jurisdictions to ensure sharks and other migratory species receive enough protection throughout their migrations, including regulations focused on highly utilized habitats in each region and movement corridors [21–24].

A recent initiative called *MigraVías* (www.migramar.org), calls for the creation of protected " swimways" and corridors between MPAs, islands and seamounts for the conservation of migratory species [22]. By linking populations throughout the seascape there is less chance of extinction and greater support for species richness and population resilience to climate change [23]. *MigraVías* must be based on scientific evidence of marine megafauna moving between insular habitats to safeguard marine corridors and maintain connectivity between islands through conservation measures [20,23]. Considerable evidence is available [1,4,17,20,22], but more studies are planned to fully justify the creation of these corridors. Scientists and national governments have worked closely to create the first two migration corridors in the ETP: 1) the Coiba-Malpelo MigraVía between Panama and Colombia (ref??), and 2) the Coco-Galapagos MigraVía between Costa Rica and Ecuador [22]. More recently, there is evidence of migration corridors between the Gulf of California and Revillagigedo and Clipperton (https://sharkrayareas.org/portfolio-item/gulf-california-revillagigedo-clipperton-corridor-isra/) that could support the creation of other MigraVías in the Mexican Pacific [17–18].

The definition of the extent and occurrence of long-range movements and population connectivity are necessary for a full understanding of the behavioral ecology of a species and hence, for designing effective conservation action [24–25]. By assessing movement frequency, network analysis (NA) can be used to identify important movement paths between core habitats of a species [25–26]. NA is a tool used to map and analyze connections between different locations or habitats, often applied in studying animal movements [26]. NA provides a new insight into the connectivity of specific habitats and the animals moving between them. It also proves valuable in revealing important information on distinct spatial and temporal changes in animal movements [5,25,26].

Movement frequency is useful for identifying important paths because frequent travel between locations often indicates critical routes for resources like food or breeding sites. However, it is typically complemented by other factors, such as habitat quality and environmental conditions, to fully understand the significance of these paths [25]. For example, an area with a high degree of centrality would suggest strong site fidelity by highly migratory animals, hence the animals may return to the same location from many different areas [5].

Movements and site fidelity patterns of top predators are still poorly understood, particularly within and between insular locations [17,24]. Pelagic sharks often exhibit site fidelity or move between islands due to biological and ecological factors. These sharks may return to specific areas for breeding, feeding, or habitat preferences, with some locations serving as critical nursery grounds for juveniles [20]. Ecologically, their movements are driven by prey availability, following schools of fish or squid concentrated by oceanic features like currents and upwellings [22]. Environmental factors such as currents, chlorophyl, temperature and oxygen levels also influence seasonal migrations [4,7]. While MPAs can protect key habitats, the high mobility of these sharks means that static MPAs may not fully encompass their range, necessitating more

dynamic management strategies. Studies highlight the importance of linking MPAs through migration corridors to enhance conservation efforts [22].

This study aims to describe the movement patterns of Galapagos sharks (*Carcharhinus galapagensis*) and silky sharks (*Carcharhinus falciformis*) in RA and the ETP. Specifically, it focuses on four main objectives:

1. Characterization of movement types: To identify and describe the various types of movements exhibited by these shark species.

2. Frequency of movements and site fidelity: To evaluate how often the sharks move and their level of attachment to specific locations over time.

3. Seasonality and diel activity patterns: To analyze how movement patterns vary with seasons and across different times of the day (diel patterns).

4. Connectivity and stepping stones: To examine how sharks connect different locations and the role of specific sites as intermediary "stepping stones" in their movements.

## Materials and methods

### Subjects of study

The silky shark is a globally distributed (40°N and 40°S) and a highly migratory species [26–29]. It is found from the surface to depths of >200 m [30,31]. It is present at seamounts and in the open ocean. Carbon (δ13 C) and nitrogen (δ15 N) isotope analysis, indicates that the species feeds in the open ocean, consuming pelagic prey at night or in the early morning [28]. One of the common preys could be the giant squid, *Dosidiscus gigas*, during its vertical migration to the surface at night [29–30].

The Galapagos shark has a similar geographical (39°N-33°S) and depth distribution (from the surface to 180 m, but mostly <80 m) to the silky shark. It is also often associated with seamounts and oceanic islands. However, it does not generally inhabit pelagic waters but is commonly found along the continental shelf [31]. Galapagos sharks feed primarily on demersal teleosts, but they also consume cephalopods, crustaceans, small marine mammals (e.g., sea lions), and even other elasmobranch species [32].

### Study area

The movements of sharks were monitored in the RA (18º49′N 112º46′W), a no-take MPA covering 148,000 km² around a group of four volcanic islands, 445 km southwest of Cabo San Lucas, Mexico. The three eastern islands, San Benedicto, Socorro, and Roca Partida, referred to as the inner islands, are relatively close to each other (<140 km apart). Clarion is 280 km to the west, and thus is called the outer island [11].

We also identified movements to and from other MPAs in the ETP. Cabo Pulmo National Park (CP) was designated as an MPA in 1995 with an extension of 71 km² and 30% of no-take area, however the no-take area has increased to nearly 100% in recent years [33]. It was declared a UNESCO World Heritage site in 2005 and a Ramsar Site in 2008 [33]. Clipperton Atoll (10°17'N 109°13'W) is positioned at the eastern edge of the Eastern Pacific Barrier (Briggs 1961; [34]), this is the only coral atoll in the region and is 965 km from mainland Mexico. The 50-m isobath is ~500 m from the reef comprising a 3.7 km² coral circle. Cocos Island (5º31'N 87º04'W) is located more than 500 km from mainland Costa Rica with an area of 24 km², and surrounded by an insular shelf that drops to a depth of 180 m, comprising a marine area of 300 km², then drops off to a depth of several thousand meters [35]. This island is the only land mass on the Cocos Ridge, which originates from the Galapagos Spreading Center. Malpelo (3°58′N and 81°37′W) is situated 490 km from the Colombian Pacific coast, with an area of 1.2 km² and surrounded by eleven pinnacles with its highest point 300 m above sea level

[36]. The Galapagos Archipelago (0º40'S 90º33'W) is located 1,000 km from the coast of Ecuador. The archipelago is made up of 13 large islands with over 100 islets and emergent rocks along with an unknown number of shallow and deep seamounts [12]. The six MPAs (CP, RA, CA, CO, MA and GA) in the ETP are characterized by their complex oceanography and high diversity and abundance of pelagic species with high economical value for fisheries and tourism [10,22].

## Fieldwork

Ultrasonic transmitters (Vemco, Ltd., Halifax; V16; frequency, 69 kHz; power, 152–158 dB re 1 µPa @ 1 m; life, 1800–3650 days) were attached externally and internally to 92 sharks, of which 39 were *Carcharhinus falciformis* and 53 were *C. galapagensis* (Table 1) during cruises to RA, CO, GA, MA from 2010 to 2018. The tagging information is given in S1 Table. The transmitters emitted a coded acoustic series of pulses of a frequency of 69 kHz with a pseudo-random delay of 60–180 s to avoid successive signal collisions between the pulse trains of two or more tags. The tags were attached externally to sharks by scuba and free diving with pole spears or spearguns by inserting a stainless-steel barb into the dorsal musculature at the base of the dorsal fin. Other tags were implanted in the peritoneal cavity of sharks caught using hook and line. The sex of each shark was determined based on the presence or absence of claspers, life stage (neonate, juvenile and adult) was recorded based on total length (TL; estimated by freedivers or measured for captured sharks) (see S1 Table).

Due to frequent hurricanes in RA during the wet season (from June to October), it was impossible to tag sharks throughout the whole year. For that reason, both species were tagged at the beginning and end of the dry season (from November to May).

An array of 50 receivers, part of the Pelagios Kakunja Ultrasonic Receiver Network (www.pelagioskakunja.org) and MigraMar Ultrasonic Receiver Network (http://www.migramar.org/), detected the signals emitted by the ultrasonic transmitters. The 50 receivers (Vemco Ltd., Halifax, VR2 and VR2W) were deployed at all five MPAs in the ETP (Table 2),

**Table 1. Tagging information of sharks monitored in the Revillagigedo National Park. Number of females and males tagged, and the maximum and minimum (cm) TL.**

|  | *C. falciformis* | *C. galapagensis* |
|---|---|---|
| Revillagigedo |  |  |
| Females | 19 (80–225 cm TL) | 9 (180–300 cm TL) |
| Males | 1 (198 cm TL) | 2 (111–250 cm TL) |
| Unknown | 1 (150 cm TL) | 7 (175–250 cm TL) |
| Clipperton |  |  |
| Female | 2 (Unknown) | 1(180 cm) |
| Galapagos |  |  |
| Females | 13 (150–224 cm TL) | 13 (140–250 cm TL) |
| Males | 2 (214–217 cm TL) | 2 (170–245 cm TL) |
| Unknown |  | 3 (100–170 cm TL) |
| Malpelo |  |  |
| Females |  | 3 (200–230 cm TL) |
| Males |  | 2 (210 cm TL) |
| Unknown |  | 1 (Unknown) |
| Cocos |  |  |
| Females | 1(226 cm) | 4 (251–300 cm TL) |
| Males |  |  |
| Unknown |  | 6 (Unknown) |

**Table 2. Receiver array information of sharks monitored in the Eastern Tropical Pacific.**

| MPA | Island | # Receivers deployed |
|---|---|---|
| Revillagigedo | Roca Partida | 3 |
| Revillagigedo | San Benedicto | 4 |
| Revillagigedo | Socorro | 3 |
| Revillagigedo | Clarion | 3 |
| Clipperton | Clipperton | 3 |
| Galapagos | Darwin | 3 |
| Galapagos | Wolf | 5 |
| Galapagos | Other islands | 16 |
| Cocos | Cocos | 5 |
| Malpelo | Malpelo | 5 |
| TOTAL | | 50 |

spanning a straight-line distance of 4000 km from Revillagigedo to the Galapagos. The arrays were deployed between 2006–2010 (Fig 1).

The receivers were affixed with heavy-duty cable ties to a mooring line leading from an anchor on the bottom (chain or concrete block) with a buoy for flotation, and positioned at depths that ranged from 12 to 43 m. The array was active during the entire monitoring period (from 2010 to 2018) and the files of tag detections were downloaded from the receivers

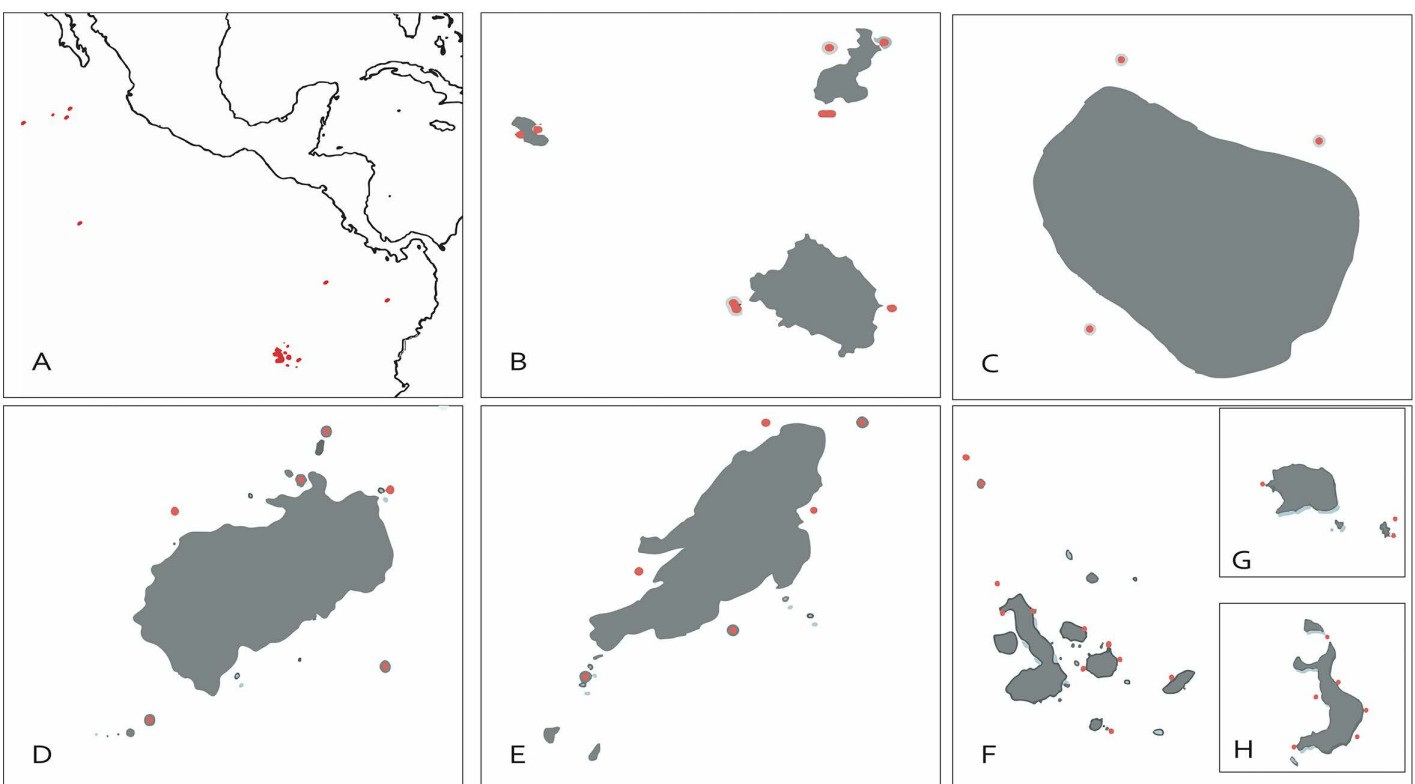

**Fig 1. Map indicating the location of acoustic receivers used to monitor shark movements in the Revillagigedo National Park (RA), Clipperton Atoll (CA), Cocos Island (CO), Malpelo Island (MA) and the Galapagos National Park (GA).**

every six months. Range tests of the ultrasonic receivers were performed at all the study sites with similar patterns across the region of detection rates >90% within 300 m, dropping to 5% at 500 m.

The work carried out in this study was done in accordance with the following research permits (resolutions) from Secretaría de Agricultura, Ganadería, Desarrollo Rural, Pesca y Alimentación and Comisión Nacional de Áreas Naturales Protegidas of Mexico: SGPA/DGVS/06798; DRPBCPN.APFFCSL-REBIARRE.-067/2011; F00.1.DRPBCPN.-00405/0216; PPF/DGOPA-134/15; PPF/DGOPA-027/14; DGOPA.03624/240413; DGOPA.06668.150612.1691; F00.DFPBCPN.000211; DGOPA.10695.191110.-5322; DGOPA.042449.270409.-1151; SGPA/DGVS/06798; DRPBCPN. APFFCSL-REBIARRE.-067/2011; F00.1.DRPBCPN.-00405/0216; PPF/DGOPA-134/15; PPF/DGOPA-027/14; DGOPA.03624/240413; DGOPA.06668.150612.1691; F00.DFPBCPN.000211; DGOPA.10695.191110.-5322; DGOPA.042449.270409.-1151. Research methods for this study were approved by the University of California, Davis Institutional Animal Care and Use Committee (IACUC) Protocol #16022.

## Data analysis

To assess the movement patterns of *Carcharhinus galapagensis* and *Carcharhinus falciformis* within and between insular sites in the RA and among insular regions in the ETP, shark movements were categorized into three distinct types: when sharks were resident at one island (behavior type I), when they moved between two or more islands within a MPA (type II), and when the sharks moved between MPAs (behavior type III). Sharks detected at a site for more than two consecutive days were considered to be present at that location. The number of acoustic receivers detecting each individual shark was recorded to evaluate the frequency of movements across different sites.

To quantify inter-island movements and dispersal ranges, the total number of receivers detecting each shark was noted, and the straight-line distances between the receivers were calculated using the geosphere library in R (v2.3.1). Frequency histograms were generated for movement distances, allowing for comparison between the two shark species.

To explore seasonality, we compiled detection data from 2010 to 2018 and segmented it into wet (June–October) and dry (November–May) seasons. These seasonal segments helped to identify patterns in movement frequency and assess the potential influence of environmental factors such as water temperature and storm activity. The seasonal variations in movement behaviors were then analyzed to determine how sharks responded to seasonal changes in the environment. The number of detections recorded for each shark during diurnal and nocturnal periods was collated separately for each receiver to examine diel patterns of activity. A Rao's test was performed to test uniformity of the daily detection patterns [37]. These differences in daytime versus nighttime detections were analyzed to explore potential foraging behaviors and habitat use at different times of the day. To evaluate site fidelity and its relationship with resource availability and protection within MPAs, the number of continuous days that individuals were resident in the study site was calculated each time they entered the study site and compared among years using one factor analysis of variance (ANOVA, 38). For all statistical analyses, the assumptions of normality and homogeneity of variance were tested using normal probability plots of residuals and plots of residuals vs. predicted values. If the data did not meet the assumptions, log transformations were performed following recommendations in Zuur [38]. Total numbers of days monitored and number of continuous day presence were calculated for each individual. Data were checked for normality with Quantile-Quantile plots and either log(x) or log (x + 1) transformed if required. Two factor analysis of variance (ANOVA), was used to test for differences in total days present and continuous days monitored between years and age classes.

Connectivity between insular sites and identification of critical locations acting stepping stones were quantified using network analysis (NA) as is explained in Leede et al. [25], utilizing the igraph package (v1.2) in R. The network was constructed by considering each location with an acoustic receiver as a node and individual shark movements between these locations as edges. Each tagged shark represented a unique observation in the network. Several key metrics were calculated to describe both the local and global structure of the network, including: (a) the number of edges, (b) the number of nodes (vertices), (c) degree of centrality, and (d) network density. Degree of centrality was used to assess the

connectedness of a node within the network, while density was calculated as the proportion of observed edges relative to all possible edges [25]. Eigenvector centrality was calculated to identify key sites within the network that were critical for shark movement and connectivity. Sites with high eigenvector centrality are those that are connected to many other highly connected sites, indicating their importance as critical hubs for shark movement and conservation efforts.

## Results

### Types of movements

A total of 92 sharks (39 silky sharks and 53 Galapagos sharks) were tagged and monitored from 2010 to 2018. Of the 92 sharks, 87 had enough detections to be considered for the analysis. Some sharks were tracked for up to five years. In general, three types of movements were identified when sharks were resident at one island (insular, type I), when they moved between two or more islands within an MPA (inter-island, type II), and when the sharks moved between MPAs (inter-regional, type III). The movements of silky sharks are described below to illustrate the different types (Fig 2):

- Silky shark #1 and #2 exhibited insular movements (Type I), detected only within Roca Partida. While #4 and #5 stayed in Socorro Island.

- Silky shark #3 exhibited inter-island movements (type II), being tagged at San Benedicto in 2012, staying for a year, traveling to Socorro, and subsequently returning to San Benedicto. It also visited Roca Partida before the detection period ended at San Benedicto in 2014.

- Silky sharks #14 and #17 showed inter-island movements (type II), traveling between San Benedicto, Roca Partida, and Socorro before returning to San Benedicto.

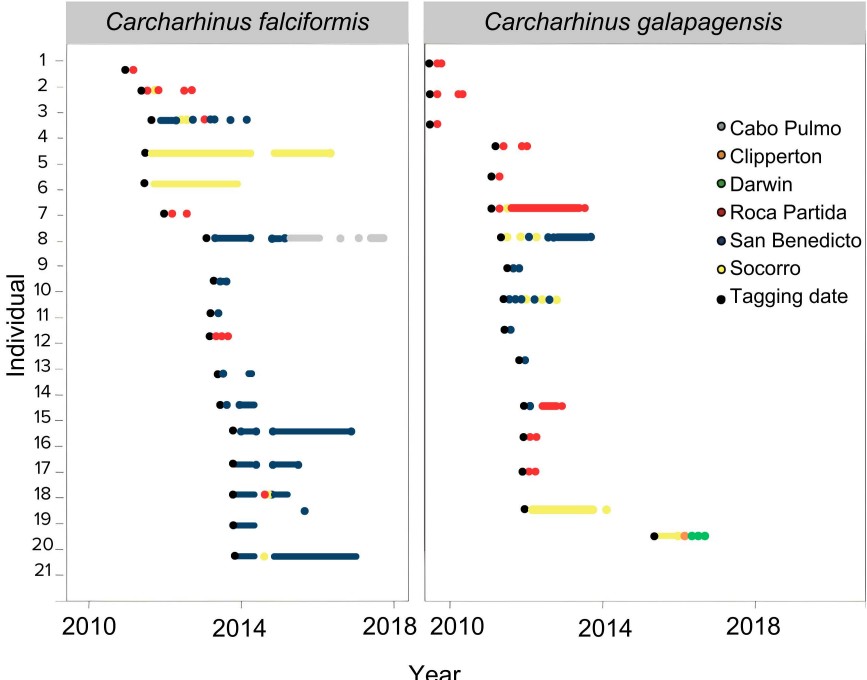

**Fig 2. Chronology of detections of *C. falciformis* and *C. galapagensis* over the last eight years in the ETP.** Islands are indicated by colors and initial tagging by a black dot.

- Silky shark #7 demonstrated inter-regional movements (type III), remaining at San Benedicto for two years before migrating to Cabo Pulmo in the Gulf of California, where it stayed until 2018.

  The Galapagos sharks also exhibited different types of movements:

- Galapagos shark #1–5 exhibited insular movement (type I). They were detected only within Roca Partida.

- Galapagos shark #6 and #7 showed inter-island movements (type II), moving from Roca Partida to Socorro, and from Socorro to San Benedicto respectively.

- Galapagos shark #15 displayed inter-regional movement (type III), traveling from Socorro (RA) to Clipperton Atoll (CA) within a year, and subsequently reaching Darwin Island (GA).

## Frequency of movements and site fidelity

From the three identified movement types (Fig 3), 65% of individuals exhibited insular movements (type I), 25% inter-island movements (type II), and 10% inter-regional movements (type III). Both species displayed high site fidelity. It reflects the degree to which an animal is associated with a particular geographic area, which may be used for activities such as foraging, breeding, resting, or shelter: Silky sharks were mostly resident for one to four months, with some individuals making brief visits lasting a day or two. Galapagos sharks tended to spend more time at Roca Partida and San Benedicto compared to Socorro.

Of the total recorded movements, 90% occurring within 50 km of the tagging site (Fig 4). Notable exceptions included: A 163 cm TL female silky shark, tagged at Roca Partida, migrating 965 km south to CA. A 187 cm TL female silky shark, tagged at Wolf Island (GA), traveling 2,200 km north to CA and returning in subsequent years. The longest recorded movement was by a 180 cm TL female Galapagos shark tagged at Socorro, which traveled 960 km south to CA and continued 2,200 km to Darwin Island (GA), completing one of the longest journeys documented for this species of 3,160 km.

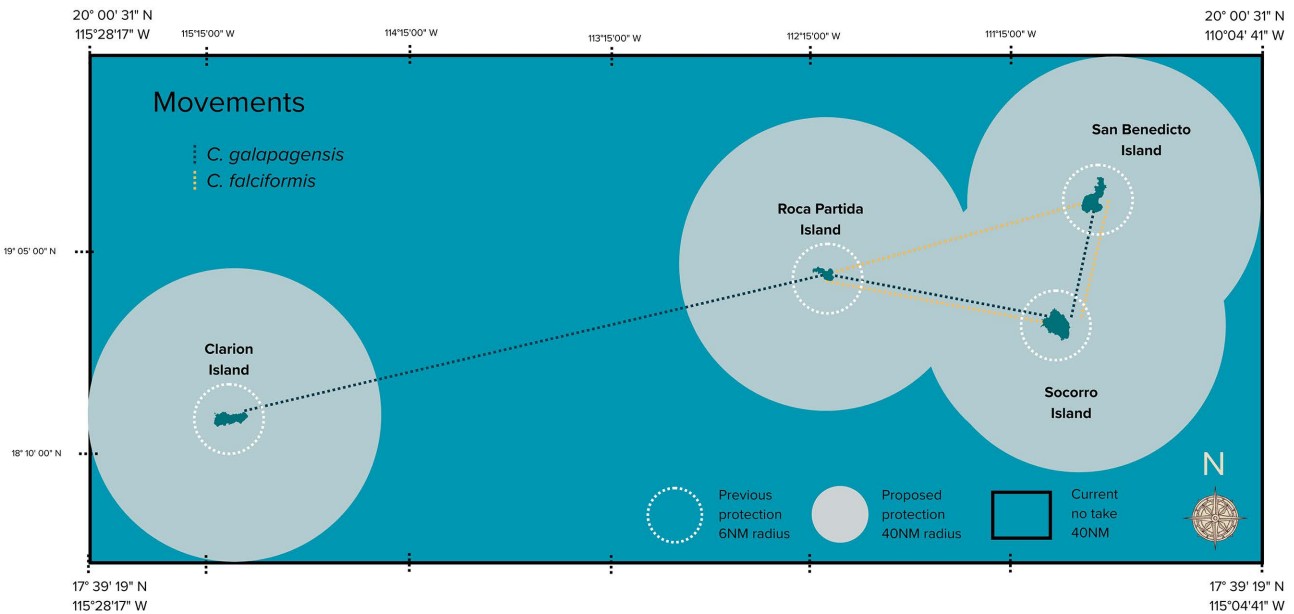

**Fig 3. Inter-Island movements of *C. galapagensis* and *C. falciformis* recorded in the RA.**

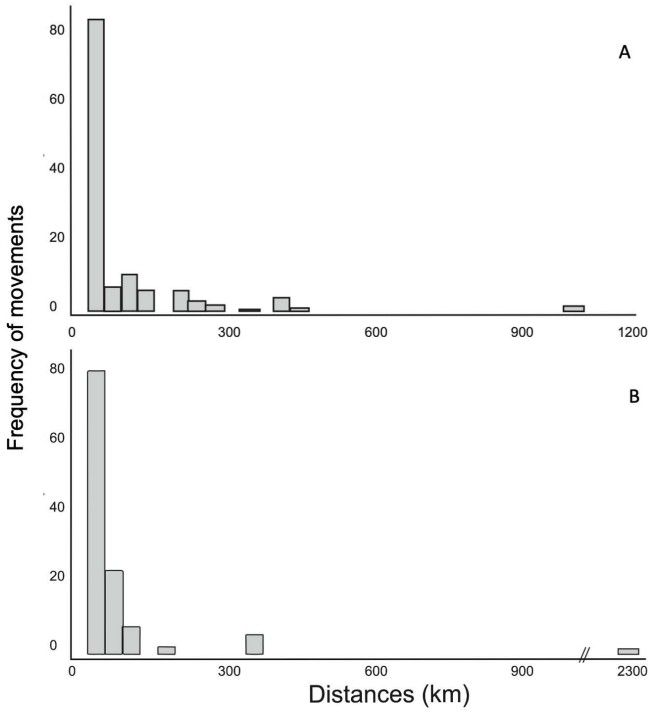

**Fig 4. Frequency of sharks' *C. falciformis* (A) and *C. galapagensis* (B) movements per distance (kilometers) in the RA.**

Both *C. galapagensis* and *C. falciformis* demonstrated a high degree of site fidelity, spending significant time at specific locations within the Archipelago. Silky sharks were primarily resident at certain sites for periods ranging from one to four months, with some individuals making brief visits lasting only a few days. Whereas, Galapagos sharks, on the other hand, displayed even higher site fidelity, especially at Roca Partida and San Benedicto, where they tended to spend extended periods compared to Socorro Island.

### Seasonality and diel patterns

Shark movements showed clear seasonal patterns. Significant differences were found between months for *C. falciformis* (f = 3.45, DF = 11, p < 0.05), and *C. galapagensis* (f = 2.35, DF = 11, p < 0.05). During the wet season (June to October), there was an increase in inter-island movements, particularly among silky sharks, which coincided with warmer water temperatures and increased storm activity. In contrast, during the dry season (November to May), sharks demonstrated higher site fidelity, with movements predominantly within a single island.

While *C. falciformis* were recorded mostly during night hours (2:00–5:00 hrs), *C. galapagensis* showed a diurnal presence with the highest records just before sunset (12:00–17:00hrs). In both cases, Rao's test of uniformity showed a significant *p-value* (p < 0.001). This means the detections during the day were not random and that some hours are more important than others. These distinct diel patterns were observed in the movements of both species. Galapagos sharks were detected more often during daytime, possibly foraging closer to shore or within the detection range of receivers. These patterns were consistent across multiple tagging sites within the archipelago.

### Connectivity and stepping stones

Silky sharks demonstrated a more complex network of movements than Galapagos sharks, as indicated by: Significant higher number of edges (X2 = 44.714, df = 1, p < 0.05), greater degree of centrality (X2 = 40.164, df = 1, p < 0.05) and higher

network density (X2 = 14.238, df = 1, p < 0.05). The number of nodes, however, did not significantly differ between species (X2 = 0.10001, df = 1, p = 0.75).

The eigenvector centrality highlighted key sites as "stepping stones" such as Roca Partida, San Benedicto, and Socorro, served as central habitats for both shark species. For silky sharks (Fig 5), the Canyon at San Benedicto was identified as a critical site with high residency times, suggesting that the availability of resources in these areas supported their prolonged stays. Other important sites were the Anchorage at Darwin (GA), and Lobster Rock (CO). For Galapagos sharks (Fig 6), they showed strong site fidelity at Roca Partida, a site known for its rich marine life, which could explain their extended presence in this location. Also, Nevera Canyon (MA), and Roca Elefante and Corales Norte (GA) were important sites for the Galapagos shark. These high-centrality nodes are indicative of areas that are well connected to other important sites and are likely serving as essential stepping stones for the sharks' movement across the ETP.

The consistent presence of sharks in these areas may indicate that MPAs offer a safe environment with stable resources, which are crucial for the sharks' biological needs. The role of MPAs in protecting sharks from fishing pressure and other anthropogenic threats was also implied by the extended residency times at these protected sites.

## Discussion

### Types of movements

Galapagos and silky sharks moved multiple times between islands in the RA. Sharks tended to be more present during the wet months (June to October) and move between sites during this period. This seasonality may be associated with

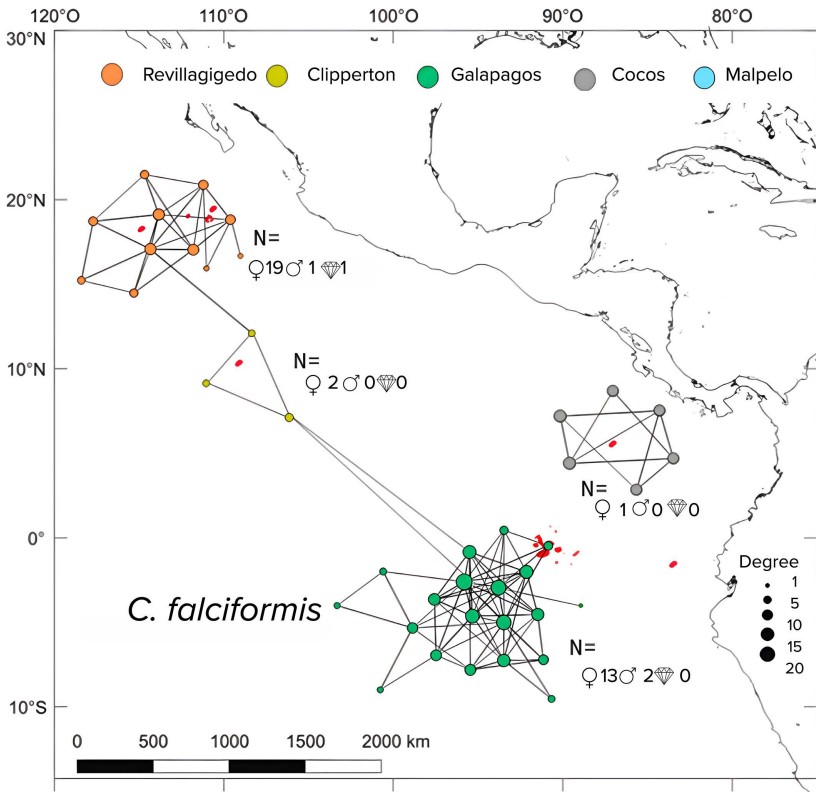

**Fig 5. Network analysis of *C. falciformis* monitored in the ETP.** Circles represent the nodes and the lines indicate the edges or movement paths. The size of the circles represents the degree, that is, the number of links for each receiver.

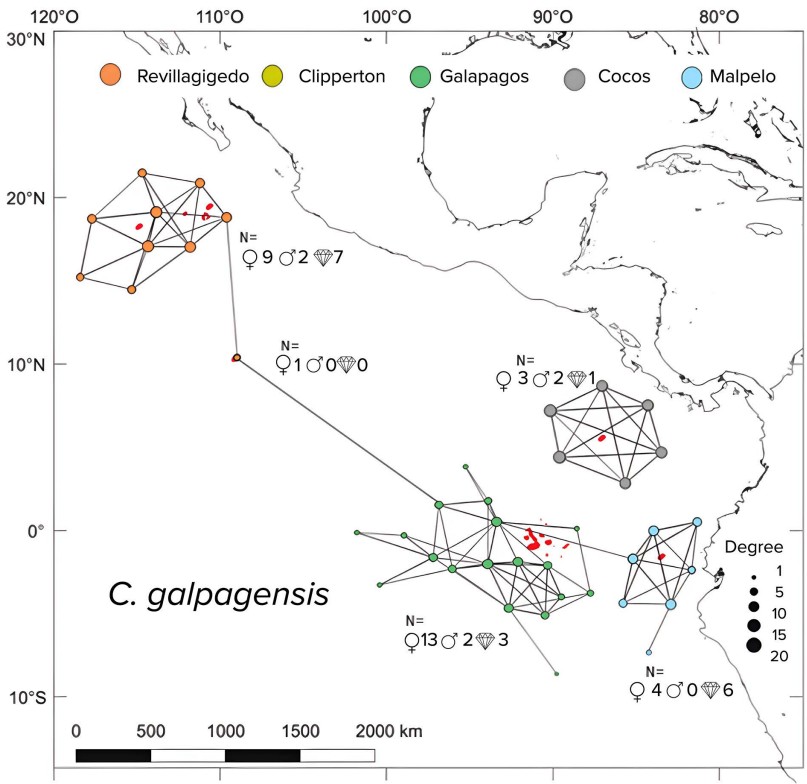

**Fig 6. Network analysis of *C. galapagensis* monitored in the ETP.** Circles represent the nodes and the lines indicate the edges or movement paths. The size of the circles represents the degree, that is, the number of links for each receiver.

tropical storms and water temperature. Other studies have shown a strong effect of sea surface temperature on the movements of scalloped hammerheads [1].

Insular movements were common in both silky and Galapagos sharks, with a clear diel separation in their detections by the nearshore receivers. This suggests silky sharks may forage offshore at night, while Galapagos sharks exhibit this behavior during the day. However, this inference must be treated cautiously due to a potential confounding variable; despite showing high fidelity to their tagging location, silky sharks also made frequent movements to adjacent sites. Comparable movement patterns are documented in related species, including silvertip sharks, whitetip reef sharks, and scalloped hammerheads [39–43].

The tracking data revealed that while both Galapagos and silky sharks are capable of moving between islands, they predominantly exhibited restricted, philopatric behavior. Most detected movements were short-range (<50 km), with a clear upper limit of around 100 km for general activity. This high site fidelity indicates that these sharks are not using a wide-ranging area within the archipelago but rather reside in core sites around specific islands. This finding is crucial for MPA design, confirming that protecting key islands provides effective protection for a significant portion of these populations.

Inter-regional movements were observed between MPAs. For example, this study has shown that Galapagos sharks can undertake long-distance movements of up to 3,160 km, from Socorro Island (RA) to Darwin Island (GA), which is the longest-ever reported movement for this species. The longest travelled distance previously recorded for a Galapagos shark (2,859 km), was performed by a male that moved off Bermuda into the central Atlantic Ocean [43]. We recorded a movement of a female Galapagos shark tagged in Socorro Island then detected southward in CA and finally detected in

Darwin Island (GA), moving 3,160 km through three distant MPAs in the ETP. Similarly, a female silky shark tagged in the Anchorage, Wolf Island (GA) travelled 2,200 km northwards to CA and back again in two different years, demonstrating inter-regional large-scale movements for this species, as other studies have shown [43].

### Frequency of movements and site fidelity

The reasons for the high fidelity could be related to the high availability of prey, and possibilities to find a mate. Also, immature sharks may be using the shallow areas around the island for refuging until they become adults [17,28]. Therefore, life stages of sharks have an important role in their space use, inter-island movements and site fidelity [40].

According to this study, the silky sharks moved more frequently to nearby sites, but also showed high fidelity to their tagging location. Some sharks may be less exposed to fishing because of the isolated location of their tagging location and/or their limited distribution within the MPAs [1,42]. Consequently, targeting specific sites based on prior knowledge and increasing the level of protection to include closely-spaced habitats (20 km) may perform better for species like *C. falciformis*, rather than having a single MPA.

Shark populations are not homogenously distributed in different habitats of the ecosystem that can support a high diversity and abundance [2,44–48]. Many shark species are known to aggregate on outer parts of reef slopes that are generally exposed to stronger current flow [5,17,48], where productive foraging grounds are present [48–49]. Hence, currents probably shape the shark community and define spatial and temporal patterns of habitat use.

Hearn *et al.* [1] and Ketchum *et al.* [17] provided evidence to support this hypothesis by showing that specific areas around Wolf Island (GA), with stronger current flow, were generally 'hotspots' for hammerhead sharks (*Sphyrna lewini*) and for other pelagic species, including Galapagos sharks. We determined that these stepping-stones are sites where earlier studies have found high abundance of sharks. Studies in CO have shown that there are fewer oceanic sharks in sheltered bays than around islands and seamounts [41–42].

Based on our study, individuals of these two species are not just highly residential, but they also start long-distance dispersal from the hotspots to other islands and MPAs. For example, Ketchum *et al.* [4] found that Dos Amigos, Roca Sucia, and Alcyone were the sites with the highest abundance for the scalloped hammerheads in CO. Manuelita also is important, but it varies according to the habitats within the site. In the RA, sharks seem to show a similar behavior as noted in the GA and MA, with the largest aggregations found up-current on the side of the island where the prevalent current flows [17]. Darwin Island (GA) may be a stopover site for scalloped hammerheads that perform long-distance movements [4].

### Connectivity and stepping stones

Based on our findings, the degree of connectivity observed among the MPAs within ETP appears to be contingent upon a limited number of individuals. Enhanced connectivity is likely to be achieved by expanding the scope of shark tracking efforts within the region. By increasing the number of sharks monitored and tracked, we can gather more comprehensive data sets that offer insights into their movements and behavior patterns. Nonetheless, disparities in receiver network deployment and acoustic coverage have notably influenced the outcomes of our analyses.

It is worth noting that our analyses did not incorporate the spatial distribution of receivers. Consequently, a higher likelihood of detecting movements over shorter distances was anticipated. Moving forward, integrating this spatial dimension into our assessments could yield a more nuanced understanding of connectivity dynamics in the ETP [47]. In sum, our research underscores the imperative for ongoing scientific inquiry and collaborative efforts aimed at the effective management and preservation of marine ecosystems in this vital region. There is an underestimation of the connectivity because some individuals can appear to be absent from receiver locations for long periods while remaining within the general study area but outside the detection range of the receivers [46]. As a critical finding, the long-distance movements recorded in this study show the potential population connectivity within the ETP.

Pazmiño *et al.* [44] used a combination of mtDNA and diagnostic nuclear markers to properly assess the genetic connectivity of the Galapagos shark across the ETP and detect patterns of hybridization. The records of hybrids of the Galapagos and dusky shark (*Carcharhinus obscurus*), showed that these species are migrating, from the RA towards the GA using CA as a stepping-stone. However, only 1% of the total sampled sharks showed this pattern. CA is an area with unusual assemblages of both Indo-Pacific and Panamic flora and fauna [15], and it is possible that it is an important stepping-stone for connecting two bioregions: northern ETP (RA, Gulf of California, [e.g., 15]) and southern ETP (MA, CO, GA).

## Conservation and management

Some sharks may gain more protection because of the location of their original site and/or the distribution of MPAs in the system [1,42]. Consequently, targeting specific sites based on prior knowledge and increasing the level of protection to include closely spaced habitats (20 km) may perform better for species like the silky shark, rather than having a single MPA [51]. Defining these movements between habitats is important to identify critical environments or corridors that may be important for population connectivity zonation [15,44] and develop management strategies that ensure protection [46].

The results of previous studies [4,17] were the rationale for expanding the old Revillagigedo Archipelago Biosphere Reserve into a rectangular area encompassing the corridors between the islands of the Revillagigedo National Park (RNP, see Fig 3). The present study as well as other recent studies further supported and justified the creation of the RNP [11]. The effectiveness of this MPA is illustrated by satellite positions of commercial fishing boats before and after the enlargement of the park (Fig 7), where occurrence of boats was substantially reduced after its creation in 2017, thus indicating a notable reduction in fishing effort for sharks and other pelagic fish (e.g., yellowfin tuna) within the RNP.

Both Galapagos and silky sharks can make long-distance movements, but how often they occur is unclear, and the shedding of externally attached tags makes it likely that tags do not stay on long enough to get infrequent long-distance migrations. It is becoming increasingly clear that some species can benefit from investments in local conservation measures nested within broader international efforts. However, Munguia *et al.* [46] and Kinney *et al.* [47] established that nursery closures or size limits that protect only neonates and young juveniles are unlikely to fully promote population recovery. That is, effective management must involve protection for older age classes along with nursery-using life stages.

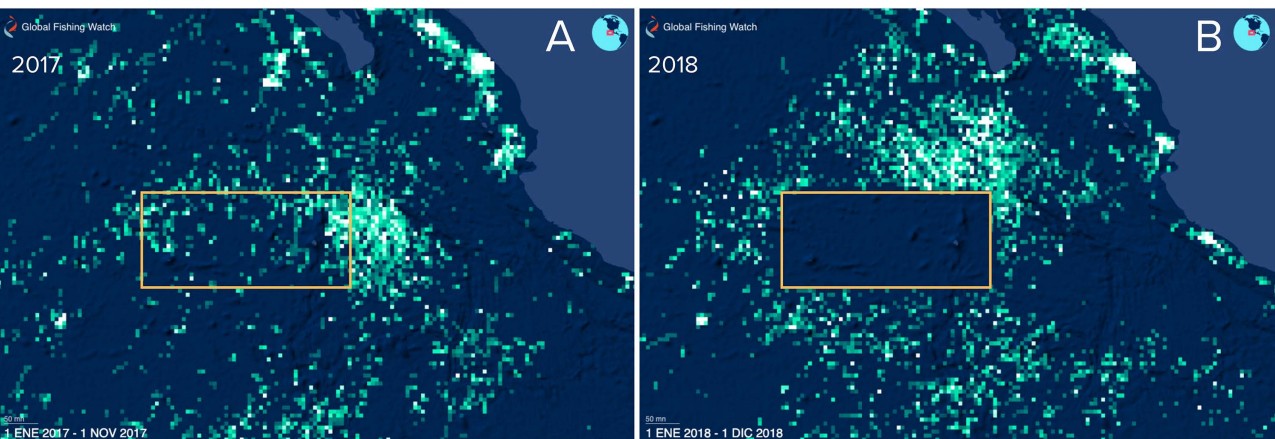

**Fig 7. Commercial Fishing effort before (A) and after (B) of the establishment of the Revillagigedo National Park (obtained from GFW, http://www.globalfishingwatch.org).**

The observed inter-regional and inter-MPAs movements suggest vulnerability to both domestic and multinational fisheries on the high seas, given their association with commercial species like yellowfin tuna, *Thunnus albacares* [29]. The preference of silky sharks to remain at or above 50 m depth makes the species much more vulnerable when moving offshore between MPAS [33]. Furthermore, even when not targeted, these sharks often comprise a high proportion of landings in line-based fisheries [48,49]. For example, Kohin *et al.* [29] determined that silky sharks tagged in Costa Rica ranged into the economic exclusive zone of six countries and beyond into international waters. Hence, definition of the extent and occurrence of long-range movement and population connectivity is necessary for a full understanding of the behavioral ecology of a species and hence for designing effective conservation action [47].

However, it has been recognized that the ETP region has a poor level of fisheries management, surveillance, and law enforcement. There is a limited ability to detect and intercept offenders, lax prosecution of legal cases, difficulties in both administrative and judicial processes, and finally, obstacles which prevent sanctions from being imposed upon offenders [10,50,51]. Therefore, the use of new technologies (i.e., remote surveillance using satellites) and international agreements should be applied more often and throughout the region.

## Conclusions

Our study sheds light on several critical aspects of shark movements within the ETP and their implications for marine conservation. First, our findings underscore the dynamic nature of connectivity between the GA and the RA, particularly for Galapagos and silky sharks. Moreover, the observed seasonality of these movements, coinciding with the wet months, suggests a potential linkage to environmental factors such as storms and water temperature.

Furthermore, our research reveals that both species undertake significant long-distance movements across the ETP, albeit such movements between distant MPAs are relatively infrequent. Nonetheless, the few instances of these large-scale movements have profound management implications, emphasizing the need for collaborative conservation efforts across MPAs.

Looking ahead, it is imperative to delve deeper into the intricacies of movement corridors and their utilization by migratory species, including conservation and management initiatives to protect these corridors such as *MigraVías* (https://migramar.org/en/migravias). Such investigations will not only enhance our understanding of the ecological dynamics within the ETP, but also provide valuable insights for the effective management and conservation of marine biodiversity in this ecologically significant region.

## Supporting information

**S1 Table. Sex, total length, time and location of tagging, monitoring duration and duration of monitoring of *C. falciformis* as well as the number of detections at islands in the eastern tropical Pacific (ETP).** San Benedicto (=SB), Socorro (=SO), Clarion (=CL), Roca Partida (RP), Clipperton = (CP), Cocos (=CO), Galapagos (=GA), and Malpelo (=MA).
(DOCX)

**S2 Table. Sex, total length, time and location of tagging, monitoring duration and duration of monitoring of *C. galapagensis* as well as the number of detections at islands in the ETP.** San Benedicto (=SB), Socorro (=SO), Clarion (=CL), Roca Partida (RP), Clipperton = (CP), Cocos (=CO), Galapagos (=GA), and Malpelo (=MA).
(DOCX)

**S3 Fig. Species comparison of network metrics of *C. falciformis* and *C. galapagensis* in the ETP.**
(DOCX)

## Acknowledgments

This study was financially supported by the International Community Foundation, Alianza WWF/Fundación Telmex-Telcel, Alianza WWF/Fundación Carlos Slim, Ocean Blue Tree, Fins Attached Marine Research and Conservation, and National Geographic-Fischer Productions. We especially thank Club Cantamar, Quino El Guardián, Rocio del Mar and Sharkwater liveaboards for providing space to travel to the islands. The authors would like to thank to Comisión Nacional de Acuacultura y Pesca, Secretaría del Medio Ambiente y Recursos Naturales, and Dirección del Parque Nacional Revillagigedo (Comisión Nacional de Áreas Naturales Protegidas) for providing necessary permits to conduct research at the Revillagigedo National Park, a UNESCO World Heritage Site. The monitoring of shark populations in Clipperton atoll was developed under a scientific collaboration agreement between the governments of France and Mexico. Dedicated to Blanca Isabel Lara Vázquez.

## Author contributions

**Conceptualization:** Frida Lara-Lizardi, James T Ketchum, Alex R. Hearn, A. Peter Klimley, Felipe Galván-Magaña, Mauricio Hoyos-Padilla.

**Data curation:** Frida Lara-Lizardi, James T Ketchum, Alex R. Hearn, A. Peter Klimley, César Peñaherrera-Palma, Mauricio Hoyos-Padilla.

**Formal analysis:** Frida Lara-Lizardi.

**Funding acquisition:** Frida Lara-Lizardi, James T Ketchum, A. Peter Klimley, Randall Arauz, Chris Fischer, Todd Steiner, Mauricio Hoyos-Padilla.

**Investigation:** Frida Lara-Lizardi, James T Ketchum, Alex R. Hearn, A. Peter Klimley, Felipe Galván-Magaña, Alex Antoniou, Randall Arauz, Sandra Bessudo, Elpis J. Chávez, Eric E.G. Clua, Eduardo Espinoza, Chris Fischer, César Peñaherrera-Palma, Todd Steiner, Mauricio Hoyos-Padilla.

**Methodology:** Frida Lara-Lizardi, James T Ketchum, Alex R. Hearn, Felipe Galván-Magaña, Elpis J. Chávez, César Peñaherrera-Palma, Mauricio Hoyos-Padilla.

**Project administration:** James T Ketchum, Felipe Galván-Magaña.

**Resources:** Frida Lara-Lizardi, Alex Antoniou, Randall Arauz, Sandra Bessudo, Eric E.G. Clua, Mauricio Hoyos-Padilla.

**Supervision:** James T Ketchum, Felipe Galván-Magaña, César Peñaherrera-Palma, Mauricio Hoyos-Padilla.

**Validation:** Frida Lara-Lizardi, Eleazar Castro, Todd Steiner.

**Visualization:** Frida Lara-Lizardi, Eleazar Castro.

**Writing – original draft:** Frida Lara-Lizardi, James T Ketchum, Alex R. Hearn, A. Peter Klimley, Randall Arauz, Sandra Bessudo, Eleazar Castro, Elpis J. Chávez, Eric E.G. Clua, César Peñaherrera-Palma, Mauricio Hoyos-Padilla.

**Writing – review & editing:** Frida Lara-Lizardi, James T Ketchum, Alex R. Hearn, A. Peter Klimley, Felipe Galván-Magaña, Elpis J. Chávez, César Peñaherrera-Palma, Mauricio Hoyos-Padilla.

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
