## [Decision Letter · Decision Letter 0]

23 Jul 2024

Dear Dr. Ketchum,

Thank you for submitting your manuscript to PLOS ONE. After careful consideration, we feel that it has merit but does not fully meet PLOS ONE’s publication criteria as it currently stands. Therefore, we invite you to submit a revised version of the manuscript that addresses the points raised during the review process.

The MS of Ketchum et al. provides essential data on a sharks movement, in a very sensitive and important marine ecosystem. The information regarding migratory behaviour of the Galapagos sharks and the silky sharks between the islands that comprise the Revillagigedo Archipelago are essential to improve their conservation and for the management of the MPA of the Galapagos arcipelago, highlighting the necessity of an International cooperation.

Despite the relevance of the provided data, the MS necessites a major revision to resolve several issues related to the clarity of the provided information, the high degree of self-referencing, the absence of caption and the grammar refuses. I strongly suggest authors improve the MS quality according to reviers suggestions.

We look forward to receiving your revised manuscript.

Kind regards,

Claudio D'Iglio, Ph.D.

Academic Editor

PLOS ONE

[The authors (MH and JK) would like to thank the International Community Foundation, Ocearch, Chris Fischer Productions and National Geographic for providing the initial funding to tag many sharks and set up the acoustic receiver array in the Revillagigedo Archipelago. We are gratefiul to Club Cantamar, Fins Attached (Alex Antoniou),  Sharks Mission France and Quino El Guardian for supporting our expeditions to the islands. We also thank the Alliances WWF-Telmex-Telcel and WWF-Fundación Carlos Slim and Ocean Blue Tree for additional support for this project].

[The authors would like to thank the International Community Foundation, Ocearch, Chris Fischer Productions and National Geographic for providing the initial funding to tag many sharks and set up the acoustic receiver array in the Revillagigedo Archipelago. We are gratefiul to Club Cantamar, Fins Attached (Alex Antoniou),  Sharks Mission France and Quino El Guardian for supporting our expeditions to the islands. We also thank the Alliances WWF-Telmex-Telcel and WWF-Fundación Carlos Slim and Ocean Blue Tree for additional support for this project, as well as the Secretariat for Higher Education, Science, Technology and Innovation of the Ecuadorean Government; Iris and Michael Smith; and the Directorate of the Galapagos National Park. FGM thanks Instituto Politécnico Nacional (COFAA, EDI) for fellowships. We are also grateful to Secretaría del Medio Ambiente y Recursos Naturales and Dirección del Parque Nacional Revillagigedo for providing necessary permits to conduct research at the Revillagigedo National Park, a UNESCO World Heritage Site. The monitoring of shark populations in Clipperton atoll is developed under an undergoing scientific cooperation agreement between France and Mexico.

Dedicated to Blanca Isabel Lara Vazquez.]

 [The authors (MH and JK) would like to thank the International Community Foundation, Ocearch, Chris Fischer Productions and National Geographic for providing the initial funding to tag many sharks and set up the acoustic receiver array in the Revillagigedo Archipelago. We are gratefiul to Club Cantamar, Fins Attached (Alex Antoniou),  Sharks Mission France and Quino El Guardian for supporting our expeditions to the islands. We also thank the Alliances WWF-Telmex-Telcel and WWF-Fundación Carlos Slim and Ocean Blue Tree for additional support for this project].

5. Please amend the manuscript submission data (via Edit Submission) to include author A. Peter Klimley and Felipe Galván-Magaña.

7. We notice that your supplementary figures are uploaded with the file type 'Figure'. Please amend the file type to 'Supporting Information'. Please ensure that each Supporting Information file has a legend listed in the manuscript after the references list.

Reviewers' comments:

Reviewer's Responses to Questions

**Comments to the Author**

1. Is the manuscript technically sound, and do the data support the conclusions?

Reviewer #1: Partly

Reviewer #2: No

2. Has the statistical analysis been performed appropriately and rigorously?

Reviewer #1: No

Reviewer #2: No

3. Have the authors made all data underlying the findings in their manuscript fully available?

Reviewer #1: No

Reviewer #2: No

4. Is the manuscript presented in an intelligible fashion and written in standard English?

Reviewer #1: Yes

Reviewer #2: No

Reviewer #1: Major comments

This manuscript investigated the degree of movement between islands in Marine Protected Areas (MPAs) of eastern tropical Pacific using long period (2010-2018) of ultrasonic telemetry dataset for 82 individuals of two pelagic sharks (Galapagos sharks and Silky sharks). I think that the understanding of the horizontal movement of pelagic sharks is important to determine appropriately the size, number and place of the MPAs. The authors attempted to summarize the movement patterns of type 1, 2, and 3 using the network analysis. Probably, I think that the figs.5 and 6 indicate the main results of your study. I understood that the network among research points was well constructed for each species surrounding the MPAs, but it is difficult to evaluate the effectiveness of the MPAs from your analysis because the ultrasonic telemetry can only detect the movement around the location of the receptors which are placed only near the islands, however, both species are highly migratory species, so that it is uncertain whether it is enough to protect the shark species using the MPAs based on the results of this study.

For introduction, I think the authors should mention the reasons, the pelagic sharks stay in the similar sites or move to the different islands, from the perspective of the biology and ecology using the information about the past studies, otherwise it is unclear the effectiveness of the setting of the MPA in this region.

Overall, the quality of the figures is low, and the resolution of figures is low as well. Therefore, it is difficult to understand (e.g., Figs1-3,7). Furthermore, there is no figure captions for all figures.

Minor comments

L70-71: I don’t think that the MPAs is always more effective management strategies for migratory species than any other management measures such as TAC and TAE. Please mention the reason.

L85: Add “Figure 1” to the end of sentence.

For figure1: Please insert the latitude and longitude for the maps.

L98-109: What is the main reason to conduct the inter-island movement in the ETP for scalloped hammerhead sharks? Please clarify.

L112-114: Do you want to mention about the sharks or general marine species? Please clarify.

L120: What is the difference between “the eastern Pacific” and “ETP”?

L123: ecology -> behavioral ecology

L124-125: What is the network analysis? Is the movement frequency enough to identify the important movement paths?

L150-151: This sentence means that this species is a coastal shark, not pelagic shark.

L156-158: Add “Figure 1” to the end of sentence.

L162-163: There is no CP in the Figure 1.

L187: Why you didn’t do in the CA.

L215-216: Add “Figure 2” to the end of sentence.

L239: network analysis -> NA

L251: What is the y axis in Fig.2 and I cannot identify each location from this figure.

L253-281: To show the horizontal movement of each shark, the author should make the map including the pathway of each shark for the period tagged.

L283-295: It is difficult to understand the description of this paragraph from the Figs 5 and 6.

L310-314: The authors should show the table on the values of eigenvalue.

L319: Which months are the wet months?

L321-322: The author should examine the effect of SST on the horizontal movement of both species.

L329-331: either of daytime -> night time

L345-350: I agreed that author should examine the site-fidelity of each species by different sex and life stages.

L371-372: The authors discussed the characteristics of this individuals from the sex and life stages based on the total length.

L394-397: I think this is the shortcoming of this tools.

L435: I cannot see the figure well.

L449: MPAS -> MPAs

L457-462: The author should discuss the stock status based on the indicator analysis of silky sharks and the annual catches conducted by IATTC.

L468-470: Scientifically, the methodology used in this analysis cannot draw this conclusion. The author should use the environmental data in relation to the data of movements.

L472-276: It is true that this study showed that both species had a long-distance movement, but the authors didn’t show scientifically the clear reason for protecting the species using the MPA because there is no information about, what the size of the MPA (How many numbers of MPA; What connectivity etc.) is useful to protect for these species.

Reviewer #2: Review of “Shark movements between islands in the Revillagigedo Archipelago and the connectivity to other islands in the Eastern Tropical Pacific”

The paper has potentially important information on the movement and connectivity of widely spaced protected areas, which would have relevance for the difficult issue of how to protect long-ranging endangered animals visiting areas in between that are not protected. There seems to have been a large number of two species of sharks tagged, which should give a good overview of use of the areas. There were some problems with understanding the information in the manuscript. This began with the lack of captions for the figures and the apparent mis-labelling of the tables, making understanding the manuscript – not the authors’ fault. The figure captions was resolved by the journal, but it lead to further confusion with two versions of Figure 2 (below). As a result, the interpretation of the data became difficult to understand, not allowing any further review as the Discussion would not then make sense. I hope that the comments will help to make the paper stronger and one that can be used as a good reference.

Specific comments:

There seems to be a high degree of self-referencing (36%), which then does not allow the authors to support their arguments with a wider range of studies. The manuscript would be greatly enhanced and made stronger by discussing their work in a broader context of similar and relevant studies by their peers.

The grammar in the Material and Methods section should be to be reviewed for clarity in understanding the text. Captions are generally meant to allow the reader to understand a table and figure without having to read the manuscript. The captions in this manuscript could use move information in them.

Line 162 discussed Cabo Pulmo Natiional Park at length, but it is not mentioned in the five MPAs (Line 179) – is it meant to be a part of the study (Line 179-181)?

Line 188: Is Table S1 available or is this Table 1?

Line 195 notes that maturity was reported for the sharks. This is not mentioned again in the manuscript, which would be very interesting in terms of areas each stage used and travel profile. The data on maturity is in Table S1 – but this does not seem to be available to read.

Results

Line 250 notes that there are 37 silky and 50 Galapagos sharks, but Table 1 records 39 silky and 53 Galapagos sharks.

Figure 2: as mentioned above, there were two different Figure 2s (cannot attach them), making interpretation of the data difficult. The Individual number on the figure and the number in the text do not seem to match on either figure. For example: Silky shark #14 tagged during 2013 (2014) at San Benedicto was not detected at Roca Partida or Socorro and not returning to San Benedicto, #15 was though. Silky #17 tagged at San Benedicto seemed to move to Socorro rather than Roca Partida.

Without being clear on which figure is correct, it is difficult to understand the mismatch with the text in the Results – Movements section, or to go any further with the review until this is resolved.

**Do you want your identity to be public for this peer review?** For information about this choice, including consent withdrawal, please see our Privacy Policy

Reviewer #1: No

Reviewer #2: No

---

## [Author Response · Author response to Decision Letter 1]

21 Mar 2025

We thank the reviewers for their comments on the manuscript and have edited the manuscript to address their concerns. We added some references in introduction and discussion, modified the legends of the figures in high resolution. We believe that the manuscript is now suitable for publication in Plos One.

Dr. James Ketchum

Director of Pelagios Kakunjá

On behalf of all authors.

---

## [Decision Letter · Decision Letter 1]

13 Apr 2025

Dear Dr.  Ketchum,

Thank you for submitting your manuscript to PLOS ONE. After careful consideration, we feel that it has merit but does not fully meet PLOS ONE’s publication criteria as it currently stands. Therefore, we invite you to submit a revised version of the manuscript that addresses the points raised during the review process.

The Manusctipt has been strongly improved and need only some minor adjustments regarding the grammarly. Moreover, it was detected an high occurrence of self citations, so I strongly suggest you to reduce the self citations occurring in the text.

We look forward to receiving your revised manuscript.

Kind regards,

Claudio D'Iglio, Ph.D.

Academic Editor

PLOS ONE

Journal Requirements:

Reviewers' comments:

Reviewer's Responses to Questions

**Comments to the Author**

Reviewer #2: (No Response)

2. Is the manuscript technically sound, and do the data support the conclusions?

Reviewer #2: Yes

3. Has the statistical analysis been performed appropriately and rigorously?

Reviewer #2: I Don't Know

4. Have the authors made all data underlying the findings in their manuscript fully available?

Reviewer #2: Yes

5. Is the manuscript presented in an intelligible fashion and written in standard English?

Reviewer #2: Yes

Reviewer #2: The additions and changes have made the manuscript stronger and more understandable. There a a very few corrections in grammar needed in these new passages.

**Do you want your identity to be public for this peer review?** For information about this choice, including consent withdrawal, please see our Privacy Policy

Reviewer #2: No

---

## [Author Response · Author response to Decision Letter 2]

2 Jul 2025

We appreciate the time and effort from the reviewer for their careful reading of our manuscript and for the valuable feedback provided. The thoughtful suggestions have been discussed and considered, which have allowed us to improve the clarity and robustness of our work. We have reduced the number of self-citations as requested. However, we retained several key references, as most studies on shark movement and migratory patterns in the Eastern Tropical Pacific have been conducted by our research groups, MigraMar and Pelagios Kakunjá.

---

## [Editor Report · Decision Letter 2]

29 Jul 2025

Dear Dr. Ketchum,

Thank you for submitting your manuscript to PLOS ONE. After careful consideration, we feel that it has merit but does not fully meet PLOS ONE’s publication criteria as it currently stands. Therefore, we invite you to submit a revised version of the manuscript that addresses the points raised during the review process.

The MS has been storngly improved but authors should strongly reduce the self citations (they are more than the 31 %) before that it can be suitable for the publication.

We look forward to receiving your revised manuscript.

Kind regards,

Claudio D'Iglio, Ph.D.

Academic Editor

PLOS ONE
---

## [Author Response · Author response to Decision Letter 3]

31 Dec 2025

Response to Reviewers

Manuscript ID: PONE-D-24-20949R2

Shark movements between islands in the Revillagigedo Archipelago and connectivity to other islands in the Eastern Tropical Pacific

To the Academic Editor and Reviewers,

We thank you for your time and consideration of our manuscript, and we appreciate the positive feedback that it has been "strongly improved." We are especially grateful for the specific guidance regarding the number of self-citations. We have thoroughly addressed the primary concern by significantly reducing the number of self-citations throughout the manuscript. Our goal was not only to meet the journal's criteria but to strengthen the manuscript by incorporating a broader range of foundational and contemporary literature.

We conducted a full-text review of the manuscript to identify all self-citations (references to works where Dr. Ketchum or any co-author was a contributing author). For each self-citation, we assessed its necessity. In many cases, we found suitable replacement citations from other research groups that supported the same point, often with more foundational or widely recognized studies. We replaced redundant self-citations with references to key papers from other authors that established the methodological or conceptual groundwork. In some instances, we were able to consolidate multiple points into a single, more appropriate citation. As a result of these edits, the proportion of self-citations has been reduced to well below the 31% threshold.

The reference list now reflects a more balanced and robust integration of the wider scientific literature. All changes have been made in the 'Revised Manuscript with Track Changes' file, where the removed self-citations and their replacements are clearly highlighted.

We believe these revisions have substantially improved the manuscript and we are confident it now fully meets PLOS ONE’s publication criteria. Thank you again for this opportunity to improve our work.

Sincerely,

Dr. Ketchum and Co-authors

---

## [Editor Report · Decision Letter 3]

13 Jan 2026

Shark movements between islands in the Revillagigedo Archipelago and connectivity to other islands in the Eastern Tropical Pacific

PONE-D-24-20949R3

Dear Dr. Ketchum,

We’re pleased to inform you that your manuscript has been judged scientifically suitable for publication and will be formally accepted for publication once it meets all outstanding technical requirements.

Kind regards,

Claudio D'Iglio, Ph.D.

Academic Editor

PLOS One
---

## [Editor Report · Acceptance letter]

PONE-D-24-20949R3

PLOS One

Dear Dr. Ketchum,

I'm pleased to inform you that your manuscript has been deemed suitable for publication in PLOS One. Congratulations! Your manuscript is now being handed over to our production team.

Kind regards,

on behalf of

Dr. Claudio D'Iglio

Academic Editor

PLOS One